# Merkel Cell Carcinoma of the Head and Neck: Epidemiology, Pathogenesis, Current State of Treatment and Future Directions

**DOI:** 10.3390/cancers13143506

**Published:** 2021-07-13

**Authors:** Mehran Behruj Yusuf, Grant McKenzie, Abbas Rattani, Paul Tennant, Jeffrey Bumpous, Donald Miller, Neal Dunlap

**Affiliations:** 1Department of Radiation Oncology, University of Louisville Hospital, Louisville, KY 40202, USA; grant.mckenzie@louisville.edu (G.M.); abbas.rattani@louisville.edu (A.R.); neal.dunlap@louisville.edu (N.D.); 2Department of Radiation Oncology, University of Alabama at Birmingham, Birmingham, AL 35294, USA; 3Department of Otolaryngology-Head and Neck Surgery and Communicative Disorders, University of Louisville Hospital, Louisville, KY 40018, USA; paul.tennant@louisville.edu (P.T.); jeffrey.bumpous@louisville.edu (J.B.); 4Department of Medicine, Division of Medical Oncology, University of Louisville Hospital, Louisville, KY 40202, USA; donald.miller@louisville.edu

**Keywords:** Merkel cell carcinoma, head and neck, immunotherapy, radiation, immunosuppression, mohs, surgery

## Abstract

**Simple Summary:**

Merkel cell carcinoma of the head and neck is a relatively uncommon cutaneous malignancy with distinct treatment management from Merkel cell carcinoma occurring in other anatomic locations. The efficacy of immunotherapy has markedly changed prognosis for patients with locally advanced or metastatic Merkel cell carcinoma of the head and neck. However, patients with primary or acquired resistance to immunotherapy remain therapeutically challenging, with novel treatment being vital to improving outcomes. Given the novel therapeutic options available for these patients, as well as increasing characterization regarding the differing oncogenesis of Merkel cell polyoma virus-positive vs. -negative tumors, up to date information regarding epidemiology; oncogenesis; current standards of treatment; and future therapeutic avenues are important to improving outcomes for patients with Merkel cell carcinoma of the head and neck.

**Abstract:**

Merkel cell carcinoma (MCC) is a rare, cutaneous neuroendocrine malignancy with increasing incidence. The skin of the head and neck is a common subsite for MCC with distinctions in management from other anatomic areas. Given the rapid pace of developments regarding MCC pathogenesis (Merkel cell polyoma virus (MCPyV)-positive or virus-negative, cell of origin), diagnosis, staging and treatment, and up to date recommendations are critical for optimizing outcomes. This review aims to summarize currently available literature for MCC of the head and neck. The authors reviewed current literature, including international guidelines regarding MCC pathogenesis, epidemiology, diagnosis, staging, and treatment. Subsequently recommendations were derived including the importance of baseline imaging, MCPyV serology testing, primary site surgery, nodal evaluation, radiotherapy, and the increasing role of immune modulating agents in MCC. MCPyV serology testing is increasingly important with potential distinctions in treatment response and surveillance between virus-positive and virus-negative MCC. Surgical management continues to balance optimizing local control with minimal morbidity. Similarly, radiotherapy continues to have importance in the adjuvant, definitive, and palliative setting for MCC of the head and neck. Immunotherapy has changed the paradigm for advanced MCC, with increasing work focusing on optimizing outcomes for non-responders and high-risk patients, including those with immunosuppression.

## 1. Background

Merkel cell carcinoma (MCC) is a rare cutaneous neuroendocrine tumor commonly affecting the skin of the head and neck with increasing incidence [1]. Often presenting as a violaceous, painless, enlarged cutaneous nodule, MCC was originally described as “trabecular carcinoma of the skin” by Dr. Cyril Toker in 1972 [2]. Much has since been elucidated regarding potential cell(s) of origin, MCC oncogenesis, clinical prognostication, and optimal treatment. MCC predominantly affects older patients (median age of diagnosis from 75 to 79 years old), with the highest incidence reported in non-Hispanic white patients, with the cutaneous tissue of the head and neck as a common anatomic subsite [3]. Men are more commonly affected by MCC relative to women, with women showing improved clinical outcomes [4]. The immune system is believed to play a critical role in MCC pathogenesis with both Merkel cell polyoma virus (MCPyV) and ultraviolet (UV) radiation identified as risk factors for oncogenesis. MCC can behave in an aggressive manner, with reported 5-year overall survival (OS) rates as low as 50.6% for patients with localized disease, though there are emerging reports of MCC patients with both local and nodal disease having more favorable disease-specific survival [5].

## 2. Oncogenesis

MCC derives its name from shared histopathologic structural and immunohistochemical characteristics with Merkel cells. Despite this, significant debate is present regarding the putative cell or cells of origin (COOs) for MCC, with epidermal stem cells [6], dermal stem cells [7], or lymphoid progenitor cells [8] postulated to be precursor cells for MCC. Advances in our understanding of MCC pathogenesis have provided further insight into MCC COOs, with potential divergent oncogenesis [9].

### Merkel Cell Polyoma Virus Positive vs. Negative MCC

In 2008, MCPyV was discovered to be associated with many (~80% in certain geographic distributions) cases of MCC [10]. MCPyV is the only known human polyoma virus associated with malignancy. MCPyV appears to be cancer promoting [11], and is clonally integrated into the host genome of MCPyV-positive MCC cells but not other cell types. Though, like other polyoma viruses, MCPyV encodes large T (LT) and small T (ST) oncoprotein antigens, MCPyV differs from other polyoma viruses in that MCPyV LT inhibits the function of RB but does not interfere with p53 [12], and the oncogenic activity of ST is attributed to inhibition of multiple E3 ligation proteins [13]. In aggregate, MCPyV oncoproteins including LT and ST inactivate p53 and RB tumor suppressors, amongst other functions [14]. This function is similar to the E6/E7 oncoproteins in human papilloma virus (HPV)-positive head and neck squamous cell carcinoma [15].

Importantly, a population of MCC is present in which no evidence of MCPyV clonal integration or expression of viral oncoproteins is present, defined as MCPyV-negative tumors. Epidemiologic data suggest the proportions of MCPyV-positive and -negative patients differ, with MCPyV-positive patients accounting for ~80% of MCC patients diagnosed in the United States, while in geographical areas with high UV exposure, such as Australia (highest worldwide incidence of MCC), MCPyV-negative MCC is predominant [16]. MCPyV-negative tumors have also been associated with increased incidence of occurring on the skin of the head and neck as opposed to the extremities [17]. Emerging data suggest important differences are present in these two MCC populations with regard to biologic behavior (MCPyV-negative tumors have been associated with increased risk of progression and MCC-specific death) [17], also paralleling HPV status for squamous cell carcinoma of the head and neck as mentioned above. Intriguingly, studies employing exome sequencing of both MCPyV-positive and -negative MCC tumors have provided insight into differing pathogenesis for these two subgroups, as well as demonstrating a striking bimodal distribution of tumor mutational burden (TMB) [18,19,20]. MCPyV-negative patients have been associated with significantly higher TMB relative to MCPyV-positive patients with MCC, with MCPyV-negative patients believed to harbor a molecular/mutation signature characteristic for UV exposure [18,20,21]. *P53/Rb1* is nearly universally mutationally inactivated in MCPyV-negative patients, with similar cellular gene mutations not consistently detected in MCPyV-positive patients, suggesting viral oncoproteins, including LT and ST, are sufficient for tumor oncogenesis/maintenance [14]. A minority of MCC tumors harbor other genomic changes such as activating events of the PI3K/AKT pathway, RAS pathway, and others [14,18,19]. Considering the above emerging data in aggregate, Sunshine and colleagues have posited that the COO for MCPyV-positive patients is present in UV-protected portions of the skin (such as the dermal fibroblasts in the dermis), with MCPyV-negative patients having a COO in a skin compartment with heavy UV exposure such as epidermal keratinocytes [9]. A simplified diagram highlighting characteristic differences between MCPyV-positive and MCPyV-negative tumors is depicted in Table 1. The clinical implications of differing biologic behavior, oncogenesis, and TMB rates between these populations may become increasingly relevant in the future with respect to improved prognostication, surveillance, development of novel targeted therapy, and improved integration of currently available immune modulating agents for the percentage of patients with primary or secondary (acquired) resistance to these patients. This is of particular significance given emerging data regarding distinct signaling pathways and viral antigen interaction for MCPyV-positive tumors that may identify novel treatment targets [22].

## 3. Clinical and Prognostic Factors

### 3.1. Extent of Disease at Presentation

Multiple baseline patient demographic, tumor, and treatment-related factors have been associated with prognosis for patients with MCC. Extent of disease at presentation (local, nodal, distant) was found to be predictive of 5-year OS in analysis of 9387 MCC cases from 1998 to 2012 abstracted from the National Cancer Database (NCDB) [3] and helped form the American Joint Committee on Cancer (AJCC) 8th edition staging for MCC. With regard to localized disease, improved outcomes were associated with pathologic confirmation of negative regional nodal basin involvement relative to clinical assessment. Clinically detected nodal involvement was also associated with worsened outcomes compared to clinically occult nodal disease. In general, larger or more locally invasive (invading fascia, cartilage, muscle, or bone) primary tumors were associated with decreased OS, with the caveat of considering tumor–node–metastasis (TNM) staging overall for prognostication instead of tumor (T) stage in isolation. AJCC staging and other known staging systems for MCC will be discussed in greater detail in a subsequent section.

### 3.2. Tumor Anatomic Subsite

With regard to anatomic subsite, the head and neck was the most common location in the overall cohort (*n* = 6144 or 42.6%), followed by upper limb and shoulder (*n* = 3397, 23.6%), lower limb and hip (*n* = 2211, 15.3%), trunk (*n* = 1575, 10.9%), and other skin (*n* = 1087, 7.5%) [3]. Anatomic subsite distribution also appears to be associated with sex, age, and ethnicity [16] with a reduced incidence of MCC of the head and neck for black patients noted. Bhatia et al. found primary tumor site to be significantly associated with OS in an NCDB analysis of 6908 patients with stage I–III MCC [23] with increased adjusted risk of mortality with MCC of the trunk for patients with stage I and II MCC, and MCC of the head and neck for patients with stage I MCC relative to MCC of the limbs (and others, reference group). MCC of the head and neck and MCC of the trunk were also significantly associated with increased mortality hazard for patients with stage III MCC in this cohort. Tumor subsite was also associated with OS by multivariable analysis (MVA) in a number of additional NCDB analyses of patients with MCC [4,24]. Specifically for MCC of the head and neck, Smith and colleagues performed an analysis of the Surveillance, Epidemiology, and End Results (SEER) database including 2104 patients with MCC of the head and neck [25] and found MCC of the lip to be associated with worsened disease-specific survival on MVA relative to MCC of the scalp, face NOS, scalp/neck, eyelid, and external ear [26].

### 3.3. Sex

Multiple studies have identified more favorable outcomes including improved OS and cancer-specific survival for women with MCC relative to men, adjusting for other relevant baseline patient, tumor, and treatment factors [4,23,24,27]. It has not yet been fully elucidated what potential underlying differences in tumor or host biology may be driving disparate outcomes in MCC by biologic sex. Given the importance of the host immune system in the development and treatment of MCC, Tam et al. posited that differences in innate and adaptive immunity by biologic sex may at least in part account for improved cancer-specific outcomes for women with MCC [4]. Intriguingly, retrospective cohort studies with increased granularity regarding immune status [28,29] have not found biologic sex to be associated with disease-specific survival on MVA, adjusting for immune status.

### 3.4. Immune Status

The immune system is believed to play an important role in both the development of MCC and subsequent therapeutic response. Increased risk of MCC development has been shown in immunosuppressed patient populations including, but not limited to, patients with HIV/AIDS [30], patients undergoing solid organ transplant [31,32], and patients with chronic lymphocytic leukemia [33] and non-Hodgkin lymphoma [34,35]. Further, MCC in immunosuppressed patients has been observed to sometimes regress with improvement in immune function [36,37], emphasizing the role immune surveillance in MCC development [38]. Immunosuppression has been shown in multiple studies to be a negative independent prognostic factor for patients with MCC [24,28,29,39,40]. With regard to differential impact on clinical outcomes by etiology of immunosuppression in MCC, Cook et al. [41] performed a retrospective analysis of 89 patients with non-metastatic MCC and found increased mortality risk for immunosuppressed patients with HIV/AIDS and organ transplant relative to patients with autoimmune disease (reference group). In an analysis of the NCDB, Yusuf et al. found immune status to be an independent predictor of OS for patients with MCC by MVA, and found etiology of immunosuppression to be associated with OS with the lowest 3-year OS rates for immunosuppressed patients with solid organ transplants [24]. Intriguingly, immunosuppressed patients in this cohort had increased likelihood of greater baseline nodal involvement and lymphovascular invasion (LVI) relative to immunocompetent patients. At least in part due to the subjective characterization of immunosuppression [42], immune status is not currently considered as part of AJCC 8th edition TNM staging for MCC and future investigations with cohorts of patients with MCC and granular information regarding MCC-specific clinical endpoints, and both etiology and quantitative severity of immunosuppression (both circulating lymphocyte populations and composition of the tumor-infiltrating lymphocytes in the TME), are paramount to further the existing understanding of immune status as a prognostic factor in MCC.

### 3.5. Tumor Microenvironment

The tumor microenvironment (TME) for MCC is an area of active research focus. Andea and colleagues showed tumor-infiltrating lymphocytes (stratified as present vs. absent) was associated with disease-specific survival (DSS), with improved DSS for patients with present TILs by univarable but not multivariable analysis [43]. Paulson et al. similarly found present TILs (relative to absent) to be associated with improved MCC-specific survival by univariable but not multivariable analysis in a retrospective study of 129 patients with MCC [44]. Yusuf and colleagues queried the NCDB and found both brisk and non-brisk TIL grade to be associated with decreased adjusted mortality hazard relative to absent TIL grade [45].

Efforts to strengthen the prognostic value of the TME for patients with MCC have focused on improved granularity for the morphology of the TME immunotype. Characterization of the immunophenotype has been suggested as a robust predictor of MCC clinical outcomes in multiple studies quantifying distinct immune cell types, including CD8+ [44,46,47], CD3+ [48,49], and FOXP3+ lymphocytes [46,49,50]. Walsh and colleagues also found brisk TIL density to be associated with MCPyV-positive tumors [50]. Further studies with large cohorts and granularity regarding TIL morphology and immunophenotype are necessary to validate the abovementioned findings, and offer increased understanding of the relationship between surrogate immune markers and other known clinically relevant prognostic factors for MCC.

### 3.6. MCPyV Status

As noted above, MCPyV-positive and -negative tumors appear to have disparate oncogenesis with concordantly distinct mutational profiles. Moshiri and colleagues performed quantitative pCR for MCPyV DNA for 282 MCC tumors and found MCC patients with virus-negative tumors in their cohort to be associated with increased risk of disease progression and death from MCC relative to virus-positive MCC patients, adjusting for factors including age, sex, and immunosuppression [17]. Recently, Harms et al. classified 346 MCC tumors from 300 patients for MCPyV status using immunohistochemistry (ICH), in situ hybridization (ISH) and quantitative pCR [14]. In their investigation, MCPyV-positive primary tumors were associated with longer disease-free survival and recurrence-free survival relative to virus-negative tumors by univariate and multivariate analysis. Further, distinct molecular prognostic markers were identified for MCC virus-positive and -negative tumors [14].

## 4. Baseline Workup and Imaging

Initial workup for patients with MCC should consist of a thorough history and physical exam including an evaluation of the primary tumor site for satellite lesions, palpable draining nodes, and dermal seeding [51]. Baseline serum levels may also be advisable (e.g., complete blood count, blood metabolic panel, alkaline phosphatase) [52]. Regional nodes are often involved and appear early in the disease course.

The importance of baseline imaging for both clinically node-positive and clinically node-negative patients with MCC has become increasingly pronounced [53] and is distinct from the management of cutaneous melanoma. National guidelines from the National Comprehensive Cancer Network (NCCN) and the Society for Surgical Oncology Choosing Wisely campaign recommend against routine baseline cross-sectional imaging for patients with localized cutaneous melanoma in the absence of clinical suspicion of adenopathy, with data suggesting very few (≤1%) patients with localized cutaneous melanoma are upstaged with baseline imaging.

Merkel cell carcinoma appears to behave in a disparate manner, with higher rates of upstaging with baseline imaging (13.2%) for clinically node-negative patients (*n* = 492) with MCC in a study of the University of Washington MCC registry (baseline imaging defined as CT, PET-CT, or magnetic resonance imaging (MRI) of the chest/abdomen/pelvis and draining nodal bed obtained within 3 months of pathologic diagnosis) [53]. In their study, baseline imaging was associated with a high positive predictive value for MCC spread in patients with newly detected lesions (94% of patients with imaging suggesting upstaging underwent pathologic confirmation, with 88.6% of these patients with subsequent pathologically confirmed MCC spread). In their cohort, PET/CT also was associated with higher rates of upstaging relative to CT alone (16.8% vs. 6.9%, *p* = 0.0006). Concordantly, NCCN guidelines encourage baseline imaging in most cases of MCC (level 2a).

While MCC does not have radiographic pathognomonic characteristics, common imaging characteristics including a cutaneous/subcutaneous nodule as the primary lesion with focal or diffuse thickening of the associated skin, commonly associated with necrosis with calcifications rare [54]. Involved lymph nodes on CT may appear enlarged with retained or compressed adipose tissue [55]. Ultrasonography with the potential for concomitant biopsy to establish pathologic diagnosis can be useful, particularly for MCC of the head and neck [56]. MCC on MRI may be hypo- to isointense on T1-weighted sequences, and may be iso- to hyperintense on T2/fat-saturated T2-weighted images [57,58]. MCC is typically a metabolically active tumor and concordantly displays elevated FDG uptake by F18-FDG PET/CT [59,60]. MCC has been shown to often express elevated levels of somatostatin receptor (SSTR) and subsequently can display increased radiotracer uptake with radioisotopes linked to peptides binding to SSTR. These radiosotopes include gallium-68 DOTA (tetraazacyclododecane tetraacetic acid)-Tyr^3^-octreotate (DOTATATE) and may become increasingly relevant for patients with SSTR-expressing MCC in the future. Currently, imaging with diagnostic CT scans with strong consideration of PET/CT is a fundamental component of baseline staging/evaluation for patients with MCC, and is integral to post-treatment surveillance for patients felt to be at increased risk of recurrence.

### 4.1. Sentinel Lymph Node Biopsy

Sentinel lymph node biopsy (SLNB) is an important tool used for assessment of the draining regional nodal basin in patients with clinically localized MCC, as identification of node-positive disease has both management and prognostic implications. SLNB involves mapping of the drainage lymphatics of the primary tumor after tissue injection of an intradermal dye, radioactive colloid, or both, followed by surgical removal of the lymphatic tissue. It is most commonly performed at the time of primary surgical resection. Once controversial in MCC management, SLNB is now recommended in both NCCN and the European Organisation for Research and Treatment of Cancer (EORTC)/European Association of Dermato-Oncology (EADO) consensus guidelines and has been incorporated into the most recent 8th edition update of the AJCC staging manual for MCC [61]. Lymphatic drainage patterns vary widely by anatomic subsite, with institutional experiences reporting particularly complex drainage patterns within the head and neck region, resulting in non-localization of sentinel nodes and false-negative biopsies [62,63]. In instances of questionable nodal drainage patterns, preoperative lymphoscintigraphy may provide additional information of the draining lymphatic basin to further improve diagnostic yield [64].

Several variables have been associated with an increased probability of a positive SLNB in MCC evaluation, including tumor size, tumor thickness, anatomic subsite, mitotic rate, infiltrate tumor growth pattern, lymphovascular invasion, presence of tumor-infiltrating lymphocytes, and immunosuppression [65,66,67]. Estimates of a positive SLNB in clinically node-negative patients vary based on institutional reports and range from 26–35% [68,69]. Presently, the prognostic value of SLNB in MCC patients has yet to be fully defined. While Kachare et al. first reported a 5.4% 5-year MCC-specific survival advantage in patients receiving SLNB compared with nodal observation in a SEER database analysis [70], prospective confirmation is lacking. A more recent multi-institutional retrospective report by Straker et al. suggests failure to detect regional nodal microscopic disease with SLNB is associated with a survival detriment, with a 5-year OS of 69.9% for patients with true negative biopsies compared to 48.1% in false negative biopsies [71]. Despite the unknown effect of sentinel node evaluation on overall survival, SLNB should be considered for all clinically node-negative MCC patients whenever possible per NCCN and EORTC/EADO guidelines, as no specific tumor characteristics have consistently been identified that portend a lower risk of a negative nodal involvement.

### 4.2. Serology

As mentioned in the above section regarding MCC oncogenesis, LT and ST antigen portions of MCPyV are both present and critical for tumor growth for ~80% of patients with MCC [72]. Paulson and colleagues demonstrated that antibodies recognizing LT or ST antigens are present in ~50% of patients with MCC, but are almost never (<1%) present in patients without MCC despite common exposure to MCPyV. Further, T-antigen antibodies decreased rapidly after effective therapy (surgical resection, radiation, or systemic therapy) for patients expressing baseline antibody levels. Seropositivity (detectable levels of T-antigen antibodies in serum) was more likely in MCC cases, consistent with MCPyV positivity. T-antigen antibody titer levels were concordant with disease burden (higher in patients with advanced disease). Serial measurement of T-antigen antibody titers also showed increased titers at time of progression for patients with disease known to have progressed. A subsequent validation study performed by the authors at the University of Washington demonstrated that seropositivity was present in 52% of patients (114 of 219 patients) with MCC in their cohort, with seropositivity at baseline was found to be a predictor of decreased recurrence risk, adjusting for age, sex, stage, and immunosuppression [73]. With regard to patients with seropositivity at baseline who underwent serial T-antigen antibody serum measurement, a decreasing oncoprotein titer had a negative predictive value of 97% for clinically evident recurrence. Given the above, baseline serology is recommended if feasible for all newly diagnosed MCC patients with strong consideration for serial oncoprotein titer assessment for seropositive patients as part of optimal surveillance. Given the association with increased risk of disease recurrence for seronegative patients, escalation of surveillance, such as more frequent diagnostic imaging, may be beneficial.

## 5. Staging

The unique histological pattern and relative aggressiveness of MCC compared to other non-melanoma carcinomas inspired an exclusive staging and coding system for MCC [74]. Appropriate staging of MCC is essential because prognosis and survival are correlated with staging at the time of diagnosis [75,76]. The initial staging system first adopted in 2010 by the American Joint Committee on Cancer (AJCC) and the Union for International Cancer Control was derived from over 2800 MCC patients with follow-up and complete staging abstracted from the NCDB. While the previous AJCC staging system incorporated the survival differences between clinical (physical exam, radiographic) versus pathologic (microscopic) evaluation of local–regional lymph nodes, it did not establish separate clinical and pathological groupings. Prior to 2010, there were no consensus staging guidelines and staging was based on a few differing studies with limited numbers of patients and institutional experiences [74,77,78]. Currently, the 8th edition AJCC staging system reflects the most recent knowledge on MCC [61]—based on the abovementioned NCDB analysis of over 9300 MCC patients [3]. Namely, a clear distinction is made between clinical and pathologic stages, and removing pathological regional nodes from stages I and II following data on the presence of occult nodal metastases in small MCC primary tumors [3].

Staging is established at the time of initial presentation prior to treatment and considers lesion size, anatomic location and invasion into adjacent structures, nodal involvement and extent of invasion, and distant metastases. The local, regional, and distant classifications are based on prognostic outcomes for each category [79]—with better prognosis for confined early-stage disease. Both tumor size and the extent of anatomic invasion are associated with survival outcomes [42,80]. A novel feature of the MCC staging system is the impact of clinical versus pathologic regional lymph nodes given the propensity of MCC to metastasize to lymph nodes [81]. Hence, pathological evaluation via SLNB or dissection is important in informing MCC staging [74]. Evaluation of the sentinel nodes, even in clinically node-negative patients, plays an important role in staging given that about one-third of clinically node-negative patients may have microscopic nodal disease [82,83,84]. Current data suggest pursuing an SLNB for clinically node-negative patients, and fine needle aspiration (FNA) or core needle biopsy for clinically palpable nodes. Excisional nodal biopsy should be considered when FNA and core needle biopsies are negative [85].

The current AJCC staging system divides MCC into two categories (i.e., clinical and pathologic) and four stages informed by data on prognostic significance (Table 2) [61]. Stage 0 represents an in situ primary tumor (Tis). Stage I is reserved for tumors ≤ 2 cm (T1) localized to the skin. Stage IIA classifies localized skin tumors >2 cm but ≤5 cm (T2), or >5 cm (T3). Stage IIB is reserved for invasive tumors beyond the skin and into fascia, muscle, cartilage, or bone (T4). Stage III and its various clinical and pathologic stage subgroups is based on degrees of nodal involvement. Clinical N1 to N3 disease represents clinical or radiographic presence of metastasis to regional nodes (N1), in-transit nodes (N2, i.e., discontinuous metastasis from primary tumor, between primary tumor or draining nodal basin, or distal to primary tumor) without nodal metastasis, or in-transit nodes with nodal metastasis (N3). Pathologic nodal classification retains the clinical N2 and N3 definitions, but with pathologically confirmed in-transit metastases. However, pathologic N1 disease is further divided into (a) clinically occult or (b) clinically/radiographically detected nodal metastasis, where pN1a(sn) and pN1a are based on identification of metastasis on SLN biopsy or dissection, respectively, and pN1b is microscopically confirmed metastasis on clinically/radiologically detected nodal metastasis. Stage III is irrespective of an identifiable/known primary tumor. This is a result of recent data suggesting that node-positive unknown primary MCC patients (T0N1b, Stage IIIA) reportedly have better prognostic outcomes compared to node-positive known primary MCC (Stage IIIB) [3,86,87]. Differentiating between known and unknown primary MCC was thus a welcome addition to the latest AJCC staging guidelines. Finally, stage IV denotes metastatic spread (M1), typically evaluated via dedicated diagnostic imaging, including PET/CT and potentially MRI of the brain if concern for intracranial metastasis is high [88]. Data are currently sparse on the impact of the M1a to M1c categories on prognosis and are thus all grouped under stage IV. M1a disease is metastasis to distant skin, subcutaneous tissue, or lymph nodes. M1b and M1c represent metastasis to the lungs or all other visceral sites, respectively. The most common site of distant metastases involves nonregional nodes, followed by liver, lung, bone, central nervous system, and other organs [55,81,89,90]. Physical exam findings, imaging, and cytology (FNA or core biopsy) may all help determine distant metastases [89].

## 6. Treatment

### 6.1. Surgical Technique

Surgical ablation of the primary tumor is recommended when feasible by national guidelines for patients with localized MCC and for selected patients with metastatic disease [85]. Primary surgical resection with the intent of obtaining histologically negative margins may be achieved with wide local excision, narrow margin excision, or with Mohs micrographic surgery (MMS) with sentinel lymph node biopsy (localized disease). Both guidelines from the NCCN and EORTC/EADO recommend 1 to 2 cm excisional margins when feasible, considering anatomy/function [85,91]. EADO/EORTC guidelines further recommend complete histologic inspection of the margins of excised tissue using microscopically controlled surgery. MMS may also be preferable in clinical situations, such as with head and neck MCC where tissue conservation may be prioritized. With regard to optimal surgical resection technique for patients with early stage MCC, the available literature is limited to retrospective single institution or database analyses in the absence of level I evidence. Shaikh et al. performed an analysis of the SEER database of patients with microscopically confirmed Merkel cell carcinoma (MCC) with 2093 patients (92.3%) treated with wide local excision and 174 patients (7.7%) treated with MMS [92]. No significant differences in OS or MCC-specific survival were demonstrated in their cohort. MMS was more likely to be used for MCC of the head and neck. Singh et al. performed an analysis of the National Cancer Database (NCDB), including 1795 patients with localized (stage I/II) MCC treated with WLE (*n* = 1685), and did not demonstrate a significant difference in OS between the two groups [93]. In the absence of prospective data, either WLE or MMS appear to be efficacious for tumor ablation for well-selected tumors. MMS in particular may be beneficial for MCC of the head and neck where tissue preservation may be paramount.

### 6.2. Surgical Margin

As mentioned above, guidelines from both the NCCN and EADO/EORTC recommend 1–2 cm resection margins in relation to the investing muscle fascia when clinically feasible [85,91]. However, there is limited clinical data with regard to the optimal resection margin for MCC. Allen and colleagues performed a retrospective analysis of 251 patients with MCC treated at Memorial Sloan Kettering Cancer Center and did not find a surgical margin of ≥1 cm to be associated with decreased local recurrence in comparison to margins <1 cm (*p* = 0.83) [77]. Perez et al. performed a retrospective examination of 240 patients with MCC treated at the Moffitt Cancer Center and did not find local recurrence to be significantly different between patients with 1 cm resection margins (2.9%), 1.1 to 1.9 cm margins (2.8%), or ≥2 cm margins (5.2%, *p* = 0.80 [94]). Of note, 69.2% of patients in their cohort received adjuvant RT. In contrast, Andruska and colleagues performed a retrospective analysis of 79 patients with stage I/II MCC treated with WLE at Washington University in St. Louis and demonstrated higher disease-specific survival for patients at 1 year with ≥2 cm margins (87.8%) relative to patients with 1 to 1.9 cm margins (71.4%) and margins < 1 cm (57.7%) [95]. The majority of patients (68%) did not receive adjuvant radiotherapy in their cohort. Additional considerations include the ability to undergo primary wound closure, which may be associated with superior cosmesis and decreased postoperative care/costs relative to graft or flap closure, and may be difficult to perform with larger resection margins (56.5% of patients in the cohort presented by Perez et al. with 1 cm margins underwent primary closure compared to 34.1% of patients with margins ≥ 2 cm [94]). When considering the available literature in total, 1 to 2 cm resection margins when clinically feasible appear appropriate with consideration for adjuvant radiotherapy in the setting of close (2 cm or less) resection margins along with additional clinical risk factors.

### 6.3. Radiation

MCC is a very radiosensitive malignancy and, concordantly, radiotherapy is commonly used as an adjuvant modality for patients with risk factors for recurrence and for definitive treatment of well-selected patients unamenable to primary surgical resection. Gillenwater et al. performed a retrospective analysis of 66 patients with MCC of the head and neck and found the use of postoperative radiotherapy to be associated with improved local and regional control, though no difference in disease-specific survival was noted [96]. Clark et al. found adjuvant radiotherapy to be associated with improved locoregional control and disease-free survival in a retrospective analysis of 110 patients with MCC of the head and neck [78]. Chen and colleagues queried the NCDB and found both postoperative RT and chemoradiation to be associated with improved OS for patients with MCC of the head and neck [97].

The NCCN suggests consideration of adjuvant RT targeting the primary tumor bed for patients with clinically node-negative localized MCC and no baseline risk factors (primary tumor > 2 cm, LVI, head and neck primary site, immunosuppression) in the setting of known clinical risk factors, including positive or close resection margins or LVI [85]. Adjuvant RT consideration of the tumor bed is recommended for patients with one or more of the aforementioned baseline risk factors in the setting of a narrow resection margin (<1 cm). Similar recommendations are given regarding primary tumor bed resection for clinically node-positive patients (without nonregional or distant disease). Immunosuppressed patients are at higher risk for recurrence [28,98], and RT should be strongly considered for these patients. Similarly, recommendations from the EORTC/EADO support consideration of adjuvant RT for most patients with cN0 MCC [91].

With regard to the draining lymph node RT, observation is recommended for most patients with localized MCC and negative SLNB with RT consideration recommended for patients at higher risk for SLNB failure (prior surgery/resection, suboptimal SLNB such as failure to perform IHC, profound immunosuppression, or with head and neck primary tumors given potential multiple draining LN basins. Of note, elective nodal RT for head and neck tumors must be carefully weighed with the potential treatment effects of irradiating multiple nodal basins). The EORTC/EADO in general does not recommend adjuvant RT of the draining nodal basin after therapeutic nodal dissection, but supports consideration of nodal RT at the discretion of a multidisciplinary tumor board, particularly in the case of involved lymph nodes with extracapsular extension [91].

With regard to time to RT initiation, the NCCN recommends expeditious initiation of RT after appropriate postsurgical healing [85]. Tsang et al. found relatively high rates of disease progression after surgery but prior to RT initiation (five of 11 patients waiting for adjuvant RT) [99]. Two recent analyses of the NCDB did not demonstrate time to RT initiation to be associated with OS for patients with localized MCC [100,101], and may offer reassurance for patients requiring additional time for optimal postsurgical healing. Given the limitations of the NCDB, including lack of cancer-specific outcomes, including locoregional control or disease progression prior to RT, attempts should be made to limit any unnecessary delays to RT initiation after wound healing.

### 6.4. Dose

Guidelines from the NCCN recommend adjuvant RT doses (conventionally fractionated at 2 Gy/fraction) of 50–56 Gy in the setting of R0 resection, 56–60 Gy in the setting of R1 resection (microscopically positive margins), and 60 to 66 Gy in the setting of R2 resection (grossly positive margins) unamenable to further resection. RT doses of 60–66 Gy are recommended for patients unamenable to surgical resection [85]. Of note, NCCN guidelines acknowledge limited evidence supporting dosing recommendations and mention that dose recommendations are provided based on clinical practice of NCCN member institutions and evidence from other cutaneous malignancies. Similar RT doses are recommended by the EORTC/EADO/EDF (50 Gy with a 10 Gy boost to tumor bed) [91]. With regard to optimal adjuvant RT dose, Patel et al. performed an analysis of the NCDB of patients with MCC of the head and neck and found adjuvant radiation doses of 50 to 55 Gy to be associated with optimal survival [102]. A subsequent analysis of the NCDB of patients with stages I–III MCC suggested that conventionally fractionated (1.8 to 2 Gy per fraction) adjuvant RT doses of 50 to 57 Gy may be associated with optimal survival for these patients [103]. Limitations inherent to analyses of the NCDB, including lack of granularity regarding radiation target/portals, and cancer-specific endpoints, including MCC-specific death and local, regional, and distant control, are present and should be considered when evaluating such investigations. Further, it is unclear if optimal adjuvant RT doses vary according to MCPyV status, which merits future investigation.

Both NCCN and EORTC/EADO/EDF guidelines recommend consideration of RT for palliation of symptomatic MCC unamenable to resection/RT as definitive treatment. Palliative dose fractionation schema include 30 Gy in 10 fractions, 20 Gy in 4 or 5 fractions, and 8 Gy in 1 fraction which can be considered for symptomatic primary, regional, and distant sites of disease. In particular, single fraction radiotherapy (8 Gy) has been demonstrated with excellent target control and favorable treatment effect profiles for patients with metastatic MCC [104], and intriguingly has been associated with durable local control and limited treatment effects in a retrospective analysis of 12 patients with localized (stage I/II) MCC of the head and neck treated with surgical resection followed by single fraction RT [105]. Such hypofractionated regimens merit further prospective study with larger patient cohorts, and currently may be reasonable to consider for patients with symptomatic metastases, or for adjuvant therapy for patients unable to receive conventionally fractionated RT.

### 6.5. Targets

With regard to optimal radiotherapy targeting of the resected primary tumor, the NCCN recommends generous (~5 cm) margins around the resected tumor bed if clinically feasible [85]. Such generous margins may be difficult to incorporate in MCC of the head and neck secondary to proximity of vital normal anatomy, and ultimate selection of radiotherapy margins must balance coverage of satellite/local in-transit disease and clinical risk factors for local recurrence (tumor size, LVSI, immune status, etc.) with treatment morbidity. Elective targeting of the in-transit lymphatics or draining regional nodes is recommended in general only when the nodal bed is in close proximity to the primary tumor. If no SLNB or nodal dissection is performed, or if conditions exist which may increase the potential for false-negative SLNB (previous WLE, operator error, failure to perform appropriate IHC on sentinel lymph nodes), elective radiotherapy of draining nodal beds can be considered. Elective nodal irradiation is also recommended to be considered in cases of unsuccessful SLNB for MCC of the head and neck, and for patients with MCC and profound immunosuppression who are at higher risk of presenting with nodal disease [24] and at higher risk for regional recurrence [106]. In a randomized trial of patients with stage I MCC treated with WLE and adjuvant RT of the tumor bed plus or minus elective nodal irradiation (ENI, 50 Gy in 25 fractions) which prematurely closed after enrolling 83 patients, ENI was associated with significantly reduced risk of regional relapse, though no significant benefit in OS was demonstrated [107]. In the setting of clinically evident adenopathy in the absence of SLNB for confirmation or therapeutic nodal dissection, radiotherapy targeting the involved nodal bed to RT doses is suggested for gross disease as noted above. Regional RT is also suggested for patients with positive SLNB or after LN dissection with multiple involved LNs or ECE. Subsites of the head and neck may have complex nodal drainage, with potential multiple draining nodal basins. Malar cheek MCC tumors may drain to ipsilateral facial nodes, the ipsilateral submandibular (IB) basin, the ipsilateral parotid nodes, or preauricular nodes, amongst other draining basins. MCC of the ear may drain to the preauricular, postauricular, or upper jugulodigastric nodes (level II), amongst other basins. Primary tumors involving the pinna or posterior to the external auditory canal may additionally drain to the posterior cervical triangle nodes (level VA/VB) [108].

Recommendations from the EORTC/EDF/EADO are similar with certain differences. SLNB is recommended whenever feasible for clinically node-negative disease, with recommendations for therapeutic nodal dissection for positive SLNB. In part due to the lack of OS benefit demonstrated in the aforementioned trial of ENI for patients with stage I MCC by Jouary et al. [107], and a lack of OS benefit demonstrated with adjuvant RT for patients with stage III (lymph node positive) MCC in an analysis of the NCDB by Bhatia et al. [23], adjuvant RT of the draining lymph nodes is not recommended in general by the European multi-society guidelines [91]. Given the paucity of level I evidence, and the differing therapeutic ratios with adjuvant radiotherapy pending anatomic subsite and other abovementioned clinical risk factors, both the NCCN and European society guidelines recommend discussion of adjuvant therapy by an MCC-specific multidisciplinary tumor board for individualized recommendations. Such tumor boards may also be optimally situated with regard to definitive radiotherapy for patients unwilling or unable to undergo primary surgical resection.

As immunotherapy becomes increasingly integrated into the treatment of metastatic, locally advanced, and localized MCC, questions remain regarding the optimal integration of radiotherapy with regard to the target (should elective nodal irradiation using traditional dose/fractionation schema be omitted given the potential for abrogation of the host immune response due to treatment related lymphopenia and radiation-stimulated cytokine signaling/pathways which may have a net immunosuppressive effect), as well as fractionation (is there a benefit to hypofractionated radiotherapy regimens which preclinically have been associated with differing effects regarding the tumor microenvironment, including immune response in other tumor subtypes [109,110,111,112,113], and are being studied in ongoing prospective clinical trials for patients with MCC (NCT03071406)). Questions also remain regarding optimal fractionation/targets for patients with profound immunosuppression who may have suboptimal clinical outcomes with conventionally fractionated RT relative to immunocompetent patients [106].

### 6.6. Immunotherapy

The advent and increasing use of immune-modulating therapies has dramatically altered the outlook for selected patients with metastatic or locally advanced MCC. The initial report of part A of the phase II JAVELIN Merkel trial of avelumab (anti-PD-L1 monoclonal antibody (mAb)) in patients with metastatic MCC with progression on prior chemotherapy demonstrated objective response in 31.8% of patients with a favorable toxicity profile [114], ultimately leading to avelumab becoming the first agent to receive FDA approval for metastatic MCC. Long-term analysis from this trial demonstrated impressive durability of response (40.5 months), with a 42-month OS rate of 31% (95% CI 22 to 41%) [115]. Assessment of PD-L1 expression in evaluable long-term survivors (LTS) (≥36 months) showed PD-L1-positive tumors in 81.8% of samples. Further, biomarker analysis of LTS demonstrated trends for improved OS and overall response rate with higher tumor mutational burden (defined as ≥2 non-synonymous somatic variants per megabase) and high major histocompatibility complex (MHC) class I expression. Grade 3 or higher treatment-related adverse events (TRAEs) were reported for 10 patients (11.4%) in this long-term update, with TRAEs leading to treatment discontinuation for eight patients [115].

Part B of the JAVELIN Merkel trial explored outcomes with Avelumab as first-line treatment for these patients, with a preplanned interim analysis demonstrating an impressive confirmed objective response rate (ORR) of 62.1% (95% CI 42.3–79.3%) [116]. Imbedded within this trial, patients enrolling in Part B were invited to participate in longitudinal qualitative phone interviews (baseline/pre-avelumab, week 13, and week 25), with data suggesting patients who self-reported disease improvement also reported perceptible improvements in physical and psychological wellbeing [117]. Pembrolizumab is an anti-PD1 mAb with demonstrated efficacy in patients with MCC [118]. Keynote-017 is a multicenter phase II trial enrolling 50 patients with naïve advanced (metastatic or recurrent locoregional disease unamenable to definitive surgery or RT) MCC, demonstrating an ORR of 56% (95% CI 41.3–70.0%). Of note, efficacy was demonstrated for both MCPyV-positive (ORR 59%) and MCPyV-negative (ORR 53%) patients, with a noted trend for improved clinical outcomes for patients with PD-L1-positive tumors [119]. The 24-month OS rate was 68.7% and median OS time was not reached with a median follow-up time of 14.9 months. Grade 3 or greater toxicities occurred in 14 of 50 patients (28%), leading to treatment discontinuation in seven patients (14%). These outcomes compare very favorably with reported outcomes with traditional therapy, with a 5-year OS rate of 13.5% reported for patients with stage IV disease [3].

Given the demonstrated efficacy for patients with advanced MCC responding to immunotherapy (IO), there is significant interest in the integration of IO for patients with localized MCC. Checkmate 358 was a phase I/II trial that evaluated clinical outcomes with preoperative nivolumab (anti-PD1 mAb) for patients with MCPyV-positive, resectable MCC [120]. Patients received IV nivolumab (240 mg) on days 1 and 15, with surgical resection planned on day 29. Thirty-nine patients with AJCC 8th edition stage IIA–IV MCC received ≥ 1 dose of nivolumab, with three patients ultimately failing to undergo resection due to progression (*n* = 1) or adverse events (*n* = 2). Grade 3 to 4 events occurred in three patients (7.7%). Of the 36 patients undergoing resection, pathologic complete response (pCR) was noted in 17 patients (47.2%). At median follow-up of 20.3 months, neither median recurrence-free survival nor OS had been reached for these patients [120]. Additional ongoing trials are investigating clinical outcomes incorporating IO for patients with localized MCC in the adjuvant setting (NCT04291885, NCT03271372, NCT03798639, NCT03712605).

Guidelines from the NCCN currently recommend consideration of immunotherapy for patients with metastatic MCC (pembrolizumab, nivolumab, or avelumab), or recurrent locally advanced MCC unamenable to definitive resection or RT (pembrolizumab) who do not have contraindications for receiving immunotherapy such as vital solid organ transplantation requiring immunosuppression or severe auto-immune conditions. Neoadjuvant or adjuvant IO off protocol is not currently recommended.

### 6.7. Novel Therapeutic Avenues/Treatment for Patients with Primary or Acquired Resistance to Immunotherapy

Despite dramatically improved outcomes for patients with advanced MCC and potentially for patients with localized MCC with primary response to IO, patients with initial or acquired resistance to currently available IO agents remain highly therapeutically challenging. As noted above, select MCC tumors demonstrate elevated SSTR expression [121]. Subsequently, there has been interest in the use of somatostatin analogues (SSAs) in the treatment of advanced MCC. Akaike et al. performed a retrospective analysis of 40 patients with metastatic MCC undergoing radiographic assessment of SSTR expression by somatostatin receptor scintigraphy (*n* = 39) or by immunohistochemistry when feasible (*n* = 9) [122]. Of the 39 patients undergoing SSTR scintigraphy, 33 patients showed some degree of uptake (low 52%, medium 23%, high 10%). Nineteen patients in this cohort were treated with an SSA analog, with seven of these patients having a target lesion amenable for response evaluation, and the remaining 12 patients having no response-evaluable lesion due to RT. The median PFS for patients treated with an SSA and an evaluable target lesion was 237 days (range 152–358 days) and 429 days (range 143–1757 days) for patients treated with an SSA without an evaluable lesion. Lutetium-177 dotatate (Lutathera) is a peptide receptor radionuclide therapy (PRRT) approved by the FDA for the treatment of SSTR-positive gastroenteropancreatic neuroendocrine tumors (GEP-NETs) based on the results of the phase III trial NETTER-1 which randomized 229 patients with well-differentiated metastatic midgut neuroendocrine tumors to either four IV infusions of Lutathera (7.4 GBq) every 8 weeks plus best supportive care including an SSA or SSA alone (octreotide long-acting repeatable (LAR)) [123]. The median PFS as not reached in the Lutathera arm at time of primary analysis and was 8.4 months (95% CI 5.8 to 9.1 months) for patients treated with the SSA alone. The single center experience at the Erasmus Medical Center included 1214 patients with SSTR-positive neuroendocrine tumors, including MCC [124], with single patient reports suggesting potential treatment response with Lutathera for patients with MCC [125]. The GoTHAM trial is a phase Ib/II trial investigating the safety and antitumor activity of Lutathera or RT in addition to avelumab for patients with metastatic MCC. Future studies investigating clinical outcomes with Lutathera for patients with both advanced and localized MCC with primary or acquired resistance to immunotherapy are vital to improving outcomes for this challenging patient population.

Efforts to improve outcomes for patients with MCC with primary or acquired resistance to immunotherapy have included targeting pathways facilitating immune escape. One such mechanism is the downregulation of major histocompatibility complex I (MHC1) expression with subsequent impairment of T cell recognition of neoplastic MCC cells [126]. Histone deacetylase class I inhibitors (HDACis) have been demonstrated to epigenetically reverse downregulation of MHC1 expression [127]. MERKLIN2 is an ongoing clinical trial exploring dominostat (HDACi) in addition to avelumab in patients with advanced MCC progressing on anti-PD(L)1 treatment (NCT04393753).

Song and colleagues performed a preclinical analysis of the efficacy and mode of action of dominostat for MCC [128]. Their results suggest dominostat can upregulate antigen-processing machinery, resulting in increased MHC1 expression. Further, functional assays in their study suggested dominostat induced both G2/M cell cycle arrest and apoptosis. Intriguingly, HDACis may both increase immunogenicity for patients with MCC and increase tumor radiosensitivity via the aforementioned G2/M cell cycle arrest. The administration of radiotherapy for MCC patients with progression on IO has been associated with immune rescue [35], and as mentioned in the section above regarding radiotherapy, is being investigated in ongoing clinical protocols (NCT03071406, NCT03304639, NCT03071406). Future investigations studying the efficacy of immunotherapy, HDACis, and RT for selected patients with advanced MCC and primary or acquired resistance to immunotherapy are needed to help further define optimal therapy for these patients.

## 7. Surveillance/Aftercare

Heterogeneity is present with regard to optimal surveillance after treatment for patients with MCC. Guidelines from the NCCN recommend physical examination, including complete skin and nodal evaluation every 3 to 6 months for the first 3 years, and every 6–12 months thereafter [85]. Imaging is recommended for patients deemed to be at increased risk of recurrence. EORTC/EADO/EDF guidelines also highlight the limited data guiding optimal surveillance. Clinical examination, including consideration of U/S examination of at-risk nodal basins, is recommended every 4 months for the first 3 years, then every 6 months for 5 years [91]. Imaging with either diagnostic CT or PET/CT is suggested yearly for the first five years. As mentioned in the section above regarding serology, serial measurement of T-antigen oncoprotein titers should be strongly considered for patients with baseline oncoprotein levels [73]. Patients with negative baseline oncoprotein titers may also warrant consideration of increased imaging and physical examination frequency.

## 8. Conclusions

Recent advances in the understanding of MCC epidemiology, pathogenesis, current treatment, surveillance, and avenues for novel therapy have significantly altered the management and outlook for patients with MCC. Surgical resection, RT, and IO are fundamental components of treatment for well-selected patients with MCC of the head and neck. Given the lack of level I evidence, relative rarity of incidence, and complexity of the native anatomy, patients with MCC of the head and neck in particular benefit from personalized recommendations by an MCC-specialized multidisciplinary tumor board composed of dermatologists, surgeons, medical oncologists, radiation oncologists, and head and neck-specialized radiologists. Future efforts tailored towards improving clinical outcomes for patients with contraindications to immunotherapy, or primary or acquired immunotherapy resistance, are vital to further improve outcomes for patients with MCC of the head and neck.

## Figures and Tables

**Table 1 cancers-13-03506-t001:** General MCC patient and tumor characteristics stratified by MCPyV status.

General Characteristics	MCPyV-Positive	MCPyV-Negative
Geographic Distribution	United States and Europe	Australia and New Zealand
Common Primary Site(s)	Head and neck, trunk, extremities	Sun-exposed skin including extremities
Tumor Mutation Burden	Lower	Higher
Molecular Oncogenesis	Viral oncoproteins including LT and ST	UV exposure, mutational inactivation of *P53/RB*
Proposed Cell of Origin	Dermal fibroblast	Epidermal keratinocyte
Tumor Behavior *	Less aggressive	More aggressive

Abbreviations: MCPyV = Merkel cell polyoma virus, UV = Ultraviolet, ST = Small T antigen, LT = Large T antigen, *: The authors acknowledge discrepancy in the available literature regarding MCPyV status as a prognostic marker, with further investigation paramount to better defining the association between viral status and tumor behavior in MCC.

**Table 2 cancers-13-03506-t002:** American Joint Committee on Cancer Staging overview for Merkel cell carcinoma.

Stage	Primary Tumor	Regional Lymph Nodes	Metastasis
Clinical			
0	Tis	N0	M0
I	T1	N0	M0
IIA	T2	N0	M0
IIB	T4	N0	M0
III	T0-4	N1-3	M0
IV	T0-4	N0-3	M1
Pathologic			
0	Tis	N0	M0
I	T1	N0	M0
IIA	T2-3	N0	M0
IIB	T4	N0	M0
IIIA	T1-4	N1a(sn) or N1a	M0
IIIA	T0	N1b	M0
IIIB	T1-4	N1b-3	M0
IV	T0-4	N0-4	M1

Staging table adapted from the American Joint Committee on Cancer’s Cancer Staging Manual 8th Edition. Chicago: Spring; 2017: 549–562.

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
