# Peer review of "Merkel Cell Carcinoma of the Head and Neck: Epidemiology, Pathogenesis, Current State of Treatment and Future Directions"

_cancers, 2021, doi:10.3390/cancers13143506_

Round 1

Reviewer 1 Report

In this review, Yusuf et al., describe the current knowledge clinical aspects of MCC of the head and neck (HNC) with detailed information on staging and current treatment options. Currently, whilst specific genomic and mechanistic data on MCC of the head and neck region is lacking, the review does contain nearly enough data on the pathogenesis of this disease.

  • HPV is another virus associated with cancers of the head and neck. It would be informational to the readers to show how MCC of the head and neck may be similar or differ to HPV+ HNC, particularly as both HPV+ HNC and MCPyV+ MCC are associated with better prognosis. This should cover aspects such as PI3K/AKT, mTOR, STAT3, NFkB and associated pathways implicated in both cancers.
  • The viruses themselves different in the activation of key drivers of oncogenesis in these cancers e.g. NFkB (Abdul-Sada et al., Oncotarget, 2017; Morgan et al., Plos Pathogens, 2019). This would be interesting to add to the review to give greater insight into how current treatment options in HPV+ HNC could/could not be applied to MCPyV+ MCC of the head and neck.
  • The above points and the review in general would benefit from  diagrams/tables highlighting some of the key points from the text, particularly for pathogenesis and/or treatment options.

Author Response

The authors would like to thank reviewer #1 for their thoughtful critiques and recommendations geared towards improving the quality of the submitted manuscript under consideration. 

We have modified the text of the manuscript to better highlight the genetic/mutational differences between MCPyV positive and negative tumors including the function of viral oncoproteins ST and LT in inactivating p53 and RB tumor suppressors vs near universal mutational inactivation of p53 and RB found in virus negative MCC. We have also modified the text to highlight similarities between the E6/E7 viral oncoproteins in HPV associated HNSCCa, as well as the prognostic implications associated with virus positive and negative status for both MCC and HNSCCa. We have modified the text of the manuscript to better emphasize the emerging data suggesting improved characterization of disparate signaling pathway activation/genomic interactions between MCPyV positive and negative tumors may identify novel therapeutic targets. Table 1 depicts characterization of MCPyV positive and negative tumors. 

Reviewer 2 Report

Merkel cell carcinoma (MCC) is a neuroendocrine skin cancer with highly aggressive and invasive features. In this review article, Yusuf et al. described the epidemiology, the pathogenesis, and current development of diagnosis and treatment of MCC. The authors did a comprehensive review to describe the oncogenic characteristics of MCC and the clinical application of the prognostic factors in MCC cases. Moreover, the basic workup and treatment of MCC were introduced clearly and completely. This review is quite useful for the study of MCC therapy. Some specific opinions are listed below.

  1. Merkel cell polyomavirus (MCPyV ) and UV exposure play the main etiological roles in MCC oncogenesis, which would cause MCPyV-associated MCC and non-viral MCC. These two subsets of MCC have different pathogenic features affecting prognostic and therapeutic strategies. The authors should introduce these two subtypes separately and respectively in the pathogenic and therapeutic aspects of MCC.
  2. The tumor microenvironment of MCC has a unique pattern. The authors should introduce it.
  3. The oncogenesis of MCC could be given a distinct figure or a pathway picture.

      4. All abbreviation should be given its full name when it was used firstly in the text.

Author Response

The authors sincerely thank Reviewer #2 for their insightful and thoughtful critiques and recommendations aimed to improve the quality of the submitted manuscript under consideration. 

We have modified the text of the manuscript to more thoroughly emphasize the disparate oncogenesis of MCPyV positive and negative tumors, including potential identification of novel therapeutic targets. We have also included viral status when discussing prognostic factors for MCC. Table 1 highlights some general distinctions between MCPyV positive and negative tumors. 

The text of the manuscript has been modified to highlight the importance of the tumor microenvironment (TME) in MCC including available literature regarding the morphologic composition of the TME including tumor infiltrating lymphocytes. 

The text of the manuscript has been modified to ensure abbreviations are given in full prior to use. 

Reviewer 3 Report

It is a good review article for Merkel cell carcinoma of the head and neck. The authors had nicely summarize the cutting edge knowledge and this manuscript is also easy to read. I enjoy reading this manuscript and would like to suggest the authors to comment on the points in the follows to enrich this article.

My specific comments:
*For prognostification, a recent study reported that the MCPyV status is an independent prognostic factor for MCC (Clin Cancer Res. 2021 May 1;27(9):2494-2504). MCPyV positivity in primary tumors was associated with longer disease-specific and recurrence-free survival. Moreover, prioritized oncogene or tumor suppressor mutations were frequent in VN-MCC but rare in VP-MCC. Please comment. 
*In the 3.4. Immune Status- This reference should also be commented in this section (Ann Surg Oncol. 2021 Apr 14. doi: 10.1245/s10434-021-09944-6.)
* In the 4.1. Sentinel Lymph Node Biopsy- Failure to detect regional nodal microscopic disease by SLNB can affect survival outcomes in patients with clinically localized MCC. The impact of false negative SLNB should also be discussed (Ann Surg Oncol. 2021 Apr 22. doi: 10.1245/s10434-021-10031-z; Laryngoscope. 2021 Mar;131(3):E828-E835).
* In the 6.3 Radiation: The time to initiation of adjuvant radiation therapy also impact on survival outcomes in MCC (Pract Radiat Oncol
. Jul-Aug 2019;9(4):e372-e385). Could the authors please comment on the indications or the risk factors for adjuvant radiotherapy? Would MCPyV status be considered?
*In the 6.4. Dose: Since the MCPyV positivity in primary tumors was associated with better recurrence-free survival, it's unclear whether higher dose is associated with better outcomes for patient's with MCPyV negativity?
*In the 6.5. Targets: It would be nice to discuss about the target volume and elective nodal volume based on some anatomical sites (e.g. ear, nose, eyebrow, eyelid, scalp, philtrum, or posterior neck). 

Author Response

The authors sincerely thank Reviewer #3 for their thoughtful and meaningful recommendations and critiques towards improving the caliber of the submitted manuscript under consideration. 

1) The text of the manuscript has been modified to better emphasize the emerging data including the recommended citation elucidating the differing pathogenesis/oncogenesis of MCPyV positive and negative tumors including signaling pathways, and the data suggesting MCPyV may be an important clinical prognostic factor. 

2) The aforementioned citation is commented on in section 3.4 as a recent analysis of the NCDB characterizing immune status as a prognostic factor regarding OS for patients with MCC. 

3) The text of the manuscript in section 4.1 has been modified in accordance with the reviewer recommendation regarding the potential detrimental impact of a false negative SLNB. 

4) Section 6.3 has been amended to discuss the potential for disease progression with increased time to initiation of postoperative RT, as well as discussing 2 NCDB analyses investigating TTI of RT and OS for patients with MCC.

5) The text of the manuscript has been amended to include NCCN/EORTC/EADO guidelines for consideration of RT including consideration of risk factors such as close/positive margin, LVI, immune status. 

6) The text of the manuscript has been amended in accordance with the reviewer recommendation to highlight the paramount need to determine if differential optimal RT doses are present by MCPyV status. 

7) The text of the manuscript has been modified in accordance with the author recommendation to provide additional information regarding potential elective nodal volumes to consider based on head and neck subsite.  

Round 2

Reviewer 1 Report

In the revised manuscript, the authors have made appropriate edits based on my previous comments.

However, reference 15, used to state how MCPyV  Lt and sT are similar to HPV E6/E7 is not an appropriate reference - the authors should use a recent review on the HPV oncoproteins, such as Scarth et al., J Gen Virol, 2021.

Author Response

The authors would like to thank Reviewer 1 for their insightful recommendations regarding the aforementioned submitted manuscript under consideration. 

We have deleted reference 15 and have instead cited the recommended reference regarding HPV oncoproteins.

Reviewer 2 Report

All of the questions have been answered adequately. The quality of this review is largely improved. I think it has reached the level for publication.

Author Response

The authors would like to thank Reviewer 2 for their time and efforts towards improving the submitted manuscript under consideration. 

Reviewer 3 Report

The authors had nicely addressed all of my concerns.

Author Response

The authors would like to sincerely thank Reviewer 3 for their efforts and insights towards improving the submitted manuscript under consideration.